# Carotenoids in Drug Discovery and Medicine: Pathways and Molecular Targets Implicated in Human Diseases

**DOI:** 10.3390/molecules27186005

**Published:** 2022-09-15

**Authors:** Damilohun Samuel Metibemu, Ifedayo Victor Ogungbe

**Affiliations:** Department of Chemistry, Physics, and Atmospheric Sciences, Jackson State University, Jackson, MS 39217-0095, USA

**Keywords:** carotenoids, cancer, diabetes, drugs, ocular diseases, cardiovascular disease, neurodegenerative diseases, aging, natural products

## Abstract

Carotenoids are isoprenoid-derived natural products produced in plants, algae, fungi, and photosynthetic bacteria. Most animals cannot synthesize carotenoids because the biosynthetic machinery to create carotenoids de novo is absent in animals, except arthropods. Carotenoids are biosynthesized from two C20 geranylgeranyl pyrophosphate (GGPP) molecules made from isopentenyl pyrophosphate (IPP) and dimethylallyl pyrophosphate (DMAPP) via the methylerythritol 4-phosphate (MEP) route. Carotenoids can be extracted by a variety of methods, including maceration, Soxhlet extraction, supercritical fluid extraction (SFE), microwave-assisted extraction (MAE), accelerated solvent extraction (ASE), ultrasound-assisted extraction (UAE), pulsed electric field (PEF)-assisted extraction, and enzyme-assisted extraction (EAE). Carotenoids have been reported to exert various biochemical actions, including the inhibition of the Akt/mTOR, Bcl-2, SAPK/JNK, JAK/STAT, MAPK, Nrf2/Keap1, and NF-κB signaling pathways and the ability to increase cholesterol efflux to HDL. Carotenoids are absorbed in the intestine. A handful of carotenoids and carotenoid-based compounds are in clinical trials, while some are currently used as medicines. The application of metabolic engineering techniques for carotenoid production, whole-genome sequencing, and the use of plants as cell factories to produce specialty carotenoids presents a promising future for carotenoid research. In this review, we discussed the biosynthesis and extraction of carotenoids, the roles of carotenoids in human health, the metabolism of carotenoids, and carotenoids as a source of drugs and supplements.

## 1. Introduction

Carotenoids are synthesized in the plastids of plants’ photosynthetic apparatus and serve as vital pigments for photosynthesis, cellular repairs, and protection [1]. Carotenoids are abundant in nature, and there are over 700 structurally diverse compounds in this class of natural products [2]. Carotenoids are the most ubiquitous polyunsaturated isoprenoids distributed in plants [3]. Carotenoids have a C-40 hydrocarbon backbone. The backbone is made from eight isoprene units connected head-to-tail, except for the central unit, which has a reverse connection [4]. These tetraterpenes are produced via the dimerization of geranylgeranyl pyrophosphate in plants, algae, bacteria, fungi, aphids, and spider mites. Carotenoids are classified into (i) carotenes or hydrocarbon carotenoids, carotenoids that are made up of carbon and hydrogen atoms, and (ii) xanthophylls, carotenoids with oxygenated hydrocarbon [Figure 1] [5].

Many phytochemicals, including carotenoids, have been investigated as potential medicines for many diseases [6]. Dietary consumption of carotenoids (lycopene, beta-carotene, alpha-carotene) has been reported to reduce the chance of developing various diseases [7]. Carotenoids also play significant roles in pollination by endowing flowers and fruits with specific colors and fragrances. Furthermore, carotenoids prevent photooxidative and heat stress damage to plant cells and also help to detoxify free radicals, thereby limiting damage to vital macromolecules within the plant [8]. In addition, carotenoids help to preserve the ecosystem through their involvement in the assembly of photosystem II (PSII), a thermodynamically important step that drives the photosynthetic processes [9]. This article provides an up-to-date review of carotenoid biosynthesis and extraction, biochemical and pharmacological actions, roles in human health, and carotenoid metabolism.

## 2. Biosynthesis of Carotenoids

The biosynthetic pathway of the most abundant carotenoids in plants is illustrated in Figure 2. The colorless C40 tetraterpene carotenoid precursor, phytoene, is formed through the fusion of two C20 geranylgeranyl pyrophosphate (GGPP) molecules from isopentenyl pyrophosphate (IPP) and dimethylallyl pyrophosphate (DMAPP) via the methylerythritol 4-phosphate (MEP) route [10]. Phytoene is transformed into ζ -carotene and then into the red pigment lycopene by the enzymes ζ-carotene desaturase, phytoene desaturase, and carotenoid isomerase (CRTISO). After lycopene is formed, the route bifurcates into a number of cyclization steps catalyzed by lycopene ε-cyclase (LCYE) and lycopene β-cyclase-catalyzed (LCYB). LCYB cyclization of lycopene results in carotenoids with two β-rings (for example, β-carotene), while the harmonized action of LCYE and LCYB generates one β-ring and one ε-ring (for example, α-carotene) [11]. The ring-specific hydroxylation of oxygenated α-carotene and β-carotene results in the production of the xanthophylls [12]. The hydroxylation of α-carotene results in zeinoxanthin or α -cryptoxanthin; either can be transformed into lutein via the ring hydroxylase enzymes (CYP97C and/or CYP97A). Also, the hydroxylation of β-carotene by β-hydroxylase results in zeaxanthin and β-cryptoxanthin. Epoxidation reactions change zeaxanthin to antheraxanthin and violaxanthin. In contrast, de-epoxidation reactions change violaxanthin to zeaxanthin via antheraxanthin through the actions of violaxanthin de-epoxidase (VDE) and zeaxanthin epoxidase (ZEP). On the other hand, violaxanthin could be modified to neoxanthin in a reaction catalyzed by neoxanthin synthase and, consequently, abscisic acid (ABA) [13]. Carotenoid biosynthesis differs in archaea and bacteria from plants. Although both pathways start with GGPP and end with lycopene, bacteria utilize only one enzyme for catalysis [14].

Archaea utilizes GGPP to synthesize its carotenoids (bacterioruberin, spheroidenone, okenone, isoreniaratene). Also, cyanobacteria (photosynthetic bacteria) synthesize β-carotene, zeaxanthin, canthaxanthin, and cryptoxanthin, rather than lutein, violaxanthin, or neoxanthin, which are common in eukaryotic photosynthetic organisms. The carotenoids by microalgae are diverse. Microalgae produces β-carotene, lutein, zeaxanthin, violaxanthin, diatoxanthin, fucoxanthin, and neoxanthin from lycopene. The synthesis of specific carotenoids by alga is a function of the prevailing favorable conditions [15]. Fungi have also been known to synthesize carotenoids; these include neurosporaxanthin, torularhodin, astaxanthin, β-carotene, and lycopene [16]. Yeast species like *Xanthophyllomyces dendrorhous* and *Rhodotorula* sp. produce astaxanthin, a dietary xanthophyll [17]. Modern metabolic engineering techniques can now be used to modulate the synthesis of specific carotenoids in fungi, yeasts, bacteria, or microalgae [18].

## 3. Extraction and Isolation of Carotenoids

Extensive efforts have been made to develop more efficient extraction and isolation techniques for carotenoids. The diverse structural polarities of carotenoids pose a serious challenge to the extraction and isolation of carotenoids [19]. Exposure to light, heat, acids, and prolonged extraction times can destroy carotenoids during extraction. Organic solvents remain the preferred solvents for the extraction of carotenoids because of their hydrophobic character. To extract non-polar carotenes (esterified xanthophylls), petroleum ether, hexane, or tetrahydrofuran (THF) are recommended. The preferred solvents for the extraction of polar carotenoids include acetone, ethanol, and ethyl acetate [20]. Water, as obtained in fruits and alga cells, is typically not good for effective carotenoid extraction. Also, heat-induced dehydration, such as oven- and microwave-drying, have been reported to cause degradation and isomerization of carotenoids [21]. Therefore, samples are freeze-dried (lyophilized) to protect carotenoids from thermal degradation, although lyophilization is energy intensive and time-consuming. Dehydration remains a viable option for samples with small amounts of water. Carotenoids, in general, must be preserved as much as possible during extraction, storage, and analysis by: (i) adding a neutralizer to samples during the extraction of carotenoids to assist in neutralizing acids which were generated from the samples to stop isomerization and rearrangement. For example, to prevent the isomerization from 5,6-epoxy- to 5,8-epoxycarotenoids [22]. (ii) By adding 0.1 percent (w/v) or less of an antioxidant to the extraction solvents, such as tert-butylhydroquinone (TBHQ), butylated hydroxytoluene (BHT), pyrogallol, or ascorbyl palmitate [23]. (iii) By preventing enzymatic oxidation by minimizing the lag time between sample maceration and extraction. (iv) By ensuring the optimum temperature for carotenoid extraction is maintained. (v) By ensuring the extraction of carotenoids is in an inert environment void of oxygen. The extraction of carotenoids sometimes involves pre-treatment steps, which remove physical and cell wall barriers from samples [24]. The main goals of pre-treatment are to enhance access to the carotenoids and increase the yield of the carotenoids from the samples. Pretreatments range from chemical (acid, base, surfactants) to enzymatic or biological methods [25]. Customarily, carotenoids are extracted with organic solvents (chloroform, hexane, methanol, dichloromethane, diethyl ether, acetone). The extraction of specific carotenoids is a function of the combinations of the different organic solvents. Factors such as the polarity and chain length of the carotenoids dictate the solvent combinations [20]. As a rule of thumb, a combination of acetone and hexane is often favored for extracting polar and nonpolar carotenoids, respectively. We have previously extracted carotenoids from plant leaves using hexane/acetone [1:1 (*v*/*v*)] [26]. In a similar vein, polar and nonpolar carotenoids are commonly extracted using an acetone/ethanol/hexane combination [27]. Acetone and ethanol are preferred for efficiently extracting carotenoids when there is a significant amount of moisture in the plant materials [28]. The extraction of carotenoids is now being investigated using ionic liquids and environmentally acceptable solvents. For the efficient extraction of carotenoids, the cell wall must be disrupted. This helps the solvent to come into contact with the carotenoids within the cell and solubilize them. Some of the methods for extracting carotenoids include maceration, Soxhlet extraction, supercritical fluid extraction (SFE), accelerated solvent extraction (ASE), microwave-assisted extraction (MAE), ultrasound-assisted extraction (UAE), pulsed electric field (PEF)-assisted extraction, and enzyme-assisted extraction (EAE).

Maceration is a straightforward technique to extract natural products, especially thermolabile substances like carotenoids. However, it requires a long extraction time and the extraction efficiency is low. Due to the absence of a heating system during extraction, which inhibits the thermal deterioration of carotenoids, maceration is typically preferred to other extraction processes [29]. Unlike maceration, Soxhlet extraction is an automated technique with good extraction effectiveness. It is time-consuming and energy costly. Due to the lengthy extraction period and high extraction temperature, Soxhlet extraction increases the likelihood of thermal degradation and cis-trans isomerization of carotenoids [30,31]. Supercritical fluids like CO_2_ serve as solvents in supercritical fluid extraction (SFE). The solubility and diffusivity of supercritical fluids are akin to liquid and gas, respectively, and can dissolve numerous natural products, including carotenoids and other thermolabile compounds. Due to its high diffusion coefficient and low viscosity, it frequently produces exceptionally pure carotenoids. The five most important factors for SFE extraction of carotenoids, according to Zaghdoudi et al. [32], are extraction temperature (40–60 °C), pressure (300–400 bar), time (30–120 min), CO_2_ density (solvent strength), CO_2_ flow rate (1–5 mL min^−1^), and entrainers concentration (5–25% *v*/*v*). The yields of polar carotenoids obtained by SFE extraction are often low, as reported by Pour and co-workers [33]. However, the optimization of SFE extraction with organic modifiers (co-solvent or entrainers), such as ethanol, can increase the yield of polar carotenoids significantly [34,35]. Pressurized liquid extraction (PLE), also referred to as accelerated solvent extraction (ASE), utilizes a small amount of solvent, high temperature, and pressure. The high pressure employed in ASE forces the extracting solvent into the matrix to denature the carotenoid-binding protein and thereby improve the extraction efficiency of carotenoids [34]. The extraction solvent(s) become less viscous as the temperature rises, making it easier for the solvent to diffuse into the sample matrix. According to Zaghdoudi et al. [32], the best conditions for extracting carotenoids with various polarities were 40 °C, methanol/tetrahydrofuran (2:8, *v*/*v*), a pressure of 103 bars, and 5 min of extraction time. The extraction temperature corresponds directly with the extraction yield and inversely with the antioxidant activity [36]. There is an increase in percentage yield with polar solvent ethanol compared to non-polar hexane [28]. ASE is an effective alternative to SFE when extracting polar carotenoids [37].

Ultrasound and microwave-assisted extraction techniques can also be used for carotenoid extraction. Microwave-assisted extraction (MAE) utilizes microwave energy to generate heat in solvents in close contact with a sample, thereby partitioning the active phytochemicals from the sample matrix into the solvent [38]. In MAE, heat and mass travel in the same direction, resulting in a synergistic effect that speeds up extraction and improves the yield. Solvent-free MAE is a green technique because it uses fewer organic solvents [39,40]. MAE is a simple, quick, and cost-effective process for extracting carotenoids. It only takes a few minutes and uses a small amount of solvent. According to Saini and Keum [30], the carotenoids yield from the MAE method are generally lower when compared with those of the Soxhlet method, and an increase in the duration of MAE causes thermal degradation of carotenoids. For carotenoid extraction using the MAE method, a temperature of roughly 60 °C is ideal [39]. MAE is an economically viable method, although thermal degradation and cis-trans isomerization are common drawbacks. Adopting intermittent radiation produced higher carotenoid recovery with concomitant improvement in antioxidant activities [41]. Ultrasound-assisted extractions (UAE) uses acoustic cavitation. In order to extract metabolites efficiently and precisely through UAE, it is necessary to optimize the ultrasonic power, intensity, temperature, and density. To obtain the highest yield of zeaxanthin (11.2 mg/g) and β-carotene (4.98 mg/g) from the green microalgae *Chlorella saccharophila,* Singh et al. [42] used response surface methodology (RSM), a solvent (acetone) ratio of 67.38 μL/mg, 27.82 power percent (total power 500 W), a pulse length of 19.7 s, and an extraction time of 13.48 min. It is important to note that UAE can be implemented as a green technique. For example, using vegetable oil, Lara-Abia et al. [43] employed UAE to extract carotenoids from Carica papaya leaves. Additionally, adding ethanol as a co-solvent to vegetable oils can enhance the extraction of most polar carotenoids, especially xanthophyll esters [43,44].

Additionally, pulsed electric field (PEF) technology applies a moderate to strong electric field strength (EFS) of between 20 and 80 kV/cm to biological material (plant, animal, or microbial cells) situated between two electrodes and between 100 and 300 V/cm in batch mode [45]. Electric pulses of a high voltage are supplied via the electrodes in an electric field. The pulses compromise the cell membrane and facilitate the extraction of intracellular carotenoids [46]. Furthermore, the electric field’s intensity determines whether the cell membrane’s permeability is temporary or constant [46]. Jaeschke et al. [47] showed that a moderate electric field (MEF) and ethanol yielded about 73% of carotenoids in microalgae *Heterochlorella luteoviridis*. In the same vein, Luengo and coworkers [28] showed a positive correlation between carotenoid concentration and the antioxidant capacity of tomato peel extracts, which was unaffected by the PEF treatment.

Lastly, the extraction of carotenoids can be accomplished using enzymes in a technique referred to as enzyme-assisted extraction (EAE). Carotenoids extracted using solvents from natural sources are associated with low yield, isomerization, thermal degradation, long extraction time, and the negative impact of the organic solvents on the quality of the extract [24]. EAE is an excellent green and environmentally friendly method for carotenoid extraction. The inherent ability of enzymes to be specific and digest cell walls and membranes makes them suitable candidates for the extraction of carotenoids from biological matrices. Compared to microwave and ultrasound-assisted extraction techniques, the EAE method has a more disruptive effect on the cell walls. In addition, it is more energy efficient than microwave and ultrasound-assisted extraction methods. According to Nath et al. [48], three different carbohydrate enzymes, Viscozyme^®^ L, pectinase, and cellulase, can be used to recover carotenoids with good yields, with Viscozyme^®^ L and pectinase producing the highest yields (80–87%). Similarly, using natural hydrophobic solvents (D-limonene, turpentine, D, L-menthol, sunflower oil, and eutectic solvents) and the multi-enzyme complex Viscozyme, Ricarte and co-workers [49] recovered carotenoids from the waste of the sunflower oil, the sunflower’s petals, and florets. They concluded that enzymatic aided extraction with hydrophobic green solvents and Viscozyme produced a substantially higher recovery of carotenoids than the conventional solvent, n-hexane [49].

## 4. Carotenoids in Human Health

### 4.1. Carotenoids in Human Health: Cancer

Cancers are aberrant tissue growth and proliferation associated with the mutation of genes and disruption of normal cellular physiology [50]. Most cancers involve the reprogramming of energy metabolism and evasion of the immunological defense system [51]. According to GLOBOCAN 2020, there were about 19.3 million new cases of cancer and about 10 million mortalities in that year alone [52]. Unfortunately, most anticancer drugs have severe side effects, resistance, and toxicity. In light of this, more options for cancer treatment and management are needed. Phytochemicals have been used to treat and manage some cancers, and significant efforts are still being expended in identifying new natural product-based anticancer therapies. Some phytochemicals have anti-tumorigenesis potentials, and over 70 percent of antineoplastic drugs with diverse modes of action are derived from natural product scaffolds [53]. Many investigators have reported the anti-neoplastic potentials of carotenoids [26,54,55,56,57,58,59,60]. Kunel et al. [61] have reported that the level of carotenoids (carotene and lycopene) in the plasma of head and neck squamous cell carcinomas (HNSCCs) and oral leukoplakia patients are lower than in healthy subjects. Also, Sakhi et al. [62] showed that the levels of lutein, carotene, zeaxanthin, lycopene, and total carotenoids in the blood of HNSCC patients were attenuated when compared with healthy subjects. According to Sakhi et al. [62], the level of carotenoids in the blood of HNSCC patients after radiation correlated positively with progression-free survival (PFS). The level of carotenoids in the blood can act as an indicator of vegetable and fruit intake. A surge in the intake of vegetables and fruit can hamper the development of HNSCC and, at the same time, attenuate the rate and severity of radiotherapy-associated side effects and increase PFS [59]. Kotake-Nara and co-workers [56] have also reported that fucoxanthin and neoxanthin attenuated cell viability via the induction of apoptotic machinery in human prostate cancer cells. Neoxanthin and fucoxanthin might also have the capacity to attenuate the risk of prostate cancer [56]. Fucoxanthin (Fx) inhibits the proliferation of glioblastoma [63], colon, bladder, prostate, liver, leukemia, gastric, cervical, melanoma, osteosarcoma, breast, and lung cancer cells [64]. Reports have indicated that Fx act through different mechanisms. The mechanisms include induction of autophagy [65] and apoptosis, cell cycle arrest at G1/G0, and improvement of gap junctional intercellular communication. There is evidence that Fx exploits different signaling pathways, including Akt/mTOR, Bcl-2, SAPK/JNK, JAK/STAT, NFκB, and MAPK pathways, in its anticancer effects [66]. However, it is not clear how the effect on each pathway is integrated. It is noteworthy that carotenoids can serve as adjuvants for other anticancer agents [67].

Gloria and co-workers [57] have reported that carotenoids inhibit cell proliferation, induce cell cycle arrest, and increase apoptosis in breast cancer cell lines. Crocin and crocetin (carotenoids) obtained from saffron demonstrated antineoplastic effects in breast cancer [68]. Also, our previous studies showed that carotenoids demonstrate pro-apoptotic, anti-angiogenic, and cell cycle arrest in an animal model of breast cancer [26,58,66,69]. Β-cryptoxanthin and lycopene were reported by Lim and Wang [70] to suppress the NF-κB signaling pathway in lung cancer and prostate cancer. Β-cryptoxanthin induced apoptosis in gastric cancer models (in vitro and in vivo) [71]. Lycopene inhibits cell proliferation and apoptosis in SKOV3 ovarian cancer cells [72]. Zhang et al. [73] reported that lutein induced apoptosis and inhibited the PI3K/Akt signaling pathway in A549 lung cancer cells. Astaxanthin demonstrated anticancer activity in a PC-3 cells-derived xenograft model of prostate cancer [74]. Deinoxanthin, a carotenoid extracted from radiodurans, induced apoptosis and nuclear fragmentation in HepG2 hepatoma, HT-29 colon, and PC-3 prostate cancer cells [75]. β-carotene causes cell cycle arrest in the G2/M phase by attenuating the expression of cyclin A [76]. At the same time, lycopene hampers cell cycle progression via a reduction in the cyclin D level and the retention of p27 in cycling-cdk2, leading to the inhibition of G1/CDK activities [77]. From our previous studies, carotenoids extracted from *Spondias mombin* demonstrated angiogenic, apoptotic, and HER2 mRNA downregulation by inhibiting the VEGFR-2 kinase domain, antagonizing the X-linked inhibitor of apoptosis protein (XIAP), and inhibiting HER2 ATP kinase domain, respectively (Figure 3) [26,66,69]. While a significant number of laboratory studies have reported the anticancer activities of carotenoids, a multi-ethnic epidemiological study by Park et al. [78] showed no significant correlation between the intake of carotenoids and a reduction in the risk of colorectal cancer. However, Zhang and co-workers [79] have reported that higher intakes of carotenoids can reduce the risk of epithelial ovarian cancer.

### 4.2. Carotenoids in Human Health: Diabetes

Diabetes mellitus (DM) occurs when insulin-secreting pancreatic β- cells malfunction. The malfunction may be due to autoimmunity (type 1 diabetes mellitus) or a lack of compensation for insulin resistance (type 2 diabetes mellitus; T2DM) [80]. There are over 537 million adults living with diabetes. In 2021, a total of 6.7 million death from diabetes and diabetes-related causes were reported [81]. An average of 966 billion dollars (USD) are spent annually on diabetes. Over 2 million children and adolescents live with type 1 diabetes, and about 600 million adults are predisposed to type 2 diabetes [82,83,84,85,86]. Carotenoids are biomarkers of fruit and vegetable intake [87]. An EPIC-InterAct case-cohort study in eight European countries by Zheng et al. [88] showed that carotenoid consumption in the forms of fruits and vegetables could help prevent type 2 diabetes. According to Landon and co-workers, astaxanthin (ASX) plays a part in preventing and treating DM-associated conditions [89]. An increased consumption of carotenoids and elevated total carotenoid concentration in the serum is linked to a lower risk of type 2 diabetes [90]. It attenuates glucose tolerance, increases serum insulin levels, and increases blood glucose levels in a mouse model of diabetes [87,91]. According to Kianbakht and Hajiaghaee [92], crocin, a carotenoid found in the flowers of crocus and gardenia, demonstrated hypoglycemic effects in alloxan-induced diabetic rats. Coyne et al. [93] showed an inverse correlation between the consumption of carotene and the risk of type 2 diabetes. Lycopene ameliorates oxidative stress and improves type 2 diabetes conditions [94]. Sluijs et al. [95] showed that high beta-carotene and alpha-carotene-rich diets attenuate type 2 diabetes in humans. Carotenoids increase insulin sensitivity in type 2 diabetes [96]. Sugiura et al. [97] showed that lower levels of carotenoids in the serum correlate with an increase in insulin insensitivity. The antidiabetic effects of carotenoids are exerted downstream of the insulin receptor, including through P13K/Akt phosphorylation, the modulation of hexokinase, fructose-1,6-bisphosphatase, glucose-6-phosphatase, glycogen phosphorylase, and pyruvate kinase activities, and increase the levels of GLUT4 and insulin receptor mRNA expression [98]. Numerous studies have shown that ROS (reactive oxygen species), inflammation, and oxidative stress are all implicated in the pathophysiology of diabetes [99,100]. Over 40% of glucose is moved into the polyol pathway in the diabetic state, resulting in the shortage of NAD(P)H-dependent pathways. Excess NADH acts as a substrate for NADH oxidase to generate ROS. Advanced glycation end-products (AGEs) are Schiff bases formed from the reaction of glucose and free amino groups of proteins. AGE is known to induce oxidative stress and inflammation. Hyperglycemia resulting from diabetes leads to oxidative stress and activation of macrophages and neutrophils, producing ROS. ROS damages macromolecules (proteins, lipids, carbohydrates, and nucleic acids) and induces chronic inflammatory processes [98,99,100]. Also, there are reports that carotenoids block the movement of nuclear factor κB to the nucleus by interacting with the nuclear factor κB, thereby inhibiting the downstream release of pro-inflammatory cytokines. In the same vein, carotenoids interact with the nuclear factor erythroid 2–related factor 2 (Nfr2) pathway, enhancing its release into the nucleus and activating phase II enzymes and antioxidants. The JNK/p38 MAPK signaling pathway can also be inhibited by carotenoids [101] (Figure 4).

### 4.3. Carotenoids in Human Health: Aging

Aging is a degenerative process arising from accumulated cellular damage that results in cellular malfunctioning, tissue failure, and cell death [104]. Numerous theories of aging have been proposed. However, the mitochondrial free radical theory of aging (MFRTA) remains central to the theories of aging partly because of the various studies that have alluded to it. Although many factors contribute to the aging process, free radicals and ROS contribute immensely to aging. According to MFRTA, ROS is a byproduct of aerobic metabolism. ROS causes oxidative damage to cellular macromolecules. The telomeres theory of aging revealed that telomere length (TL) is a biomarker of aging. As a cell divides, its telomeres become shorter until it dies. ROS and inflammation can accelerate telomere shortening [105]. The anti-aging potential of carotenoids is mainly due to their ability to instigate the migration of Nrf2 to the nucleus (Figure 3). Nrf2 dictates the expression of various antioxidant and detoxification genes by binding to antioxidant response elements (AREs). Nrf2 increases the cellular defense mechanisms against inflammation and oxidative stress. Carotenoids proffer beyond protection against oxidative stress and inflammation. The report of Kidd [106] showed that astaxanthin slowed age-related functional decline [106], while Weber et al. and co-workers [107] revealed that the prevalence of age-related diseases is significantly reduced with lycopene and alpha-carotene. According to Xu et al. [108], a higher dietary intake of β-carotene was associated with longer telomere length.

### 4.4. Carotenoids in Human Health: Inflammation

Inflammation is the bedrock of many diseases, from cancer to diabetes [109]. Many studies have alluded to the anti-inflammatory potential of carotenoids. We previously showed that carotenoids from *Spondias mombin* demonstrate anti-inflammatory potentials in an animal model of breast cancer [66]. Davinelli et al. [110] have also shown that astaxanthin, a dietary carotenoid, modulates Nrf2 and NF-κB. In addition, Zhao et al. [111] have shown that astaxanthin demonstrated anti-inflammatory potentials via the inhibition of the MAPKs and NF-κB pathways. Crocin, the principal carotenoid of saffron, was shown by Ghasemnejad-Berenji, [112] to demonstrate immunomodulatory and anti-inflammatory potential in COVID-19. Β-carotene and lycopene demonstrate anti-inflammatory potentials by Kawata et al. [113], and Zheng et al. [114] confirmed that lycopene ameliorates oxidative stress and inflammation in type 2 diabetic rats. The mechanisms of the anti-inflammatory potentials of carotenoids are through the modulation of Nrf2 and NF-κB. Nrf2 induces the transcription of antioxidant and detoxifying enzymes, while NF-κB instigates the transcription of inflammatory cytokines (Figure 3) [115].

### 4.5. Carotenoids in Human Health: Ocular Diseases

Some xanthophylls are present in the *macula lutea*, the yellow spot centered on the fovea. These carotenoids are known as macular pigment carotenoids (MP). They include lutein, zeaxanthin, and meso-zeaxanthin. Lutein and zeaxanthin are distributed widely in nature and are common in the plant, while meso-zeaxanthin is a lutein metabolite and a zeaxanthin stereoisomer [116]. Humans and primates cannot synthesize carotenoids de novo but must acquire them from diets [117]. The MP carotenoids are in the fovea for several biological functions; these yellow compounds absorb blue light (the short wavelength), thereby preventing damage from long-term photo-oxidation and improving visual performance via a filtration of the short wavelength. The MP is an antioxidant centered in the fovea to ameliorate ROS and oxidative stress resulting from high oxygen levels and visible light [118,119]. MP carotenoids may stimulate normal foveal and visual growth and fight blinding childhood diseases [120]. The correlation between ocular diseases and carotenoids is established [121]. Epidemiology studies revealed that high blood concentration and high consumption of lutein and zeaxanthin decreased the risk of advanced age-related macular degeneration (AMD) [122,123]. The Age-Related Eye Disease Study 2 (AREDS 2) recommended 10 mg of lutein and 2 mg of zeaxanthin for 15 mg β-carotene daily as the standard treatment for AMD. The original formulation involving β-carotene was modified because of the tendency of β-carotene to cause lung cancer in smokers [124]. According to Hammond [125], lutein and zeaxanthin play significant roles in infants’ normal visual development. Carotenoids have been used as interventions for retinitis pigmentosa (RP) and related inherited retinal and macular degenerations. The studies by Chitchumroonchokchai [126] and Gao [127] and their co-workers showed that carotenoids halt oxidative stress and reduce damage to the lens, halting the development of cataracts. A report by Liu et al. [128] and Ma [129] and their co-workers further corroborated the protective actions of lutein and zeaxanthin against cataracts. A body of evidence showed that xanthophylls protect against diabetic retinopathy [130]. Also, xanthophylls prevent retinopathy associated with premature birth (ROP) [131]. ROS and oxidative stress are the bedrock of ocular disease [132]. Therefore, the abilities of carotenoids to modulate the Nrf2 and NFk-B pathways are primarily responsible for carotenoids’ prevention, protection, and fixation of ocular diseases [110].

### 4.6. Carotenoids in Human Health: Skin

The skin is the organ that protects the body from exogenous harmful substances. ROSs internal and external oxidative stress caused by ROS promotes skin aging, resulting in wrinkles and atypical pigmentation [133]. Exposure to ultraviolet (UV) radiation releases free radicals and ROS, which play vital roles in producing lipid radicals (L•). Lipid radicals (L•) account for the destruction of the plasma membrane. UV radiation enhances premature aging and photocarcinogenesis and contributes to photodermatoses [134]. Due to their extensive conjugated double bonds, carotenoids are excellent scavengers of free radicals and ROS. They react with free radicals to form stable intermediates and/or non-reactive products. Di Mascio et al. [135] showed that lycopene is an efficient singlet oxygen quencher. Carotenoids are excellent protectors from UV [136,137,138]. They influence skin moisture, texture, physiology, and functions and have cosmetic effects [3,138,139]. Carotenoids increase basal dermal defense against UV irradiation, provide lasting protection, and aid skin health and appearance [140]. The retinoids (derivatives of β-carotene) and their metabolites are potential treatments for dermatological conditions, including psoriasis, acne, and ichthyosis [141].

### 4.7. Carotenoids in Human Health: Neurodegenerative Diseases

Neurodegenerative diseases are characterized by gradual neuronal attrition and apoptosis, resulting in cognitive, motor dysfunction, and intellectual impairment [142]. They result from conformational changes and misfolding of proteins [143]. However, it is still unclear how the biochemical changes result in neurodegeneration, making the development of curative therapeutics for neurodegenerative diseases very challenging. Chief among neurodegenerative diseases are Parkinson’s disease, amyotrophic lateral sclerosis (ALS), a stroke, Alzheimer’s disease, and Huntington’s disease. CNS inflammation and immune activation have been implicated in the pathophysiology of neurodegenerative diseases. At the moment, there are no cures for degenerative diseases. Available therapies can only slow and/or delay the onset of the diseases. However, some carotenoids have been reported to have neuroprotective potentials [144]. Cichon et al. [145] showed that carotenoids promote synaptogenesis and neurogenesis and demonstrate neuroprotective activities. Proust-Lima et al. [146] also showed that carotenoids prevent age-related neurodegeneration. There appears to be a significant correlation between lycopene levels and cognition [147]. According to Saeedi and Rashidy-Pour [148], saffron and its major constituent crocin are promising therapeutics for improving cognition in Alzheimer’s disease (AD) and stress-related disorders. Retinoids and carotenoids are potential compounds to prevent and/or treat AD. Wani et al. [149] showed that crocetin, a carotenoid isolated from *Crocus sativus,* induced autophagy in AD. Lycopene-rich tomato in a mouse model of experimental Parkinson’s disease was reported by Honarva et al. [150] to prevent the degeneration of nigral dopaminergic neurons. The correlation between carotenoid consumption and the risk of neurodegenerative diseases is established. Reduced concentrations of carotenoids were observed in neurodegenerative diseases when compared with non- neurodegenerative diseases [151]. Carotenoids adopt several mechanisms in the fight against neurodegenerative diseases through: an enhanced antioxidant defense system via the Keap1/Nrf2 pathway, ROS reduction, suppression of the inflammatory cytokine mediating effect, and the reduction in apoptotic factors [152]. Carotenoids may also promote the clearance of amyloid-β by inducing autophagy via the STK11/LKB1-mediated AMPK pathway [153].

### 4.8. Carotenoids and Cardiovascular Diseases

Cardiovascular diseases (CVD) are the most common cause of mortality globally. For example, cardiovascular diseases accounted for over 32% of all death worldwide in 2019 [154]. CVD includes arrhythmias, aorta disease, Marfa syndrome, congenital heart disease, coronary artery disease (narrowing of the arteries), deep vein thrombosis and pulmonary embolism, a heart attack, heart failure, heart muscle disease (cardiomyopathy), heart valve disease, pericardial disease, peripheral vascular disease, rheumatic heart disease, a stroke, and vascular disease (blood vessel disease) [155]. The risk factors for CVD include smoking, a lack of exercise, a poor diet, obesity, high blood pressure, high LDL or low HDL cholesterol levels, genetic inheritance, diabetes, and age. ROS can be produced by inflammation and oxidative stress [156]. There is some evidence for the hypothesis that carotenoids promote good cardiovascular health [157,158,159]. A study by Aune et al. [154] showed that a higher dietary intake of carotenoids correlates with decreased cardiovascular disease risk. Astaxanthin is an excellent anti-CVD carotenoid, probably due to its antioxidant and anti-inflammatory properties, and its potential to modulate lipid and glucose metabolism [160,161]. Astaxanthin is a nutraceutical for CVD [162]. An increase in the intake of tomato products and lycopene supplements was shown by Cheng et al. [163] to positively affect blood lipids, blood pressure, and endothelial function. Epidemiological studies revealed that diets rich in fruits and vegetables reduce the risk of cardiovascular heart diseases (CHD) [164]. Leermakers et al. [165] showed that a higher intake of lutein is associated with excellent cardiometabolic health. A Japanese population-based follow-up study on cardiovascular disease mortality and serum carotenoid levels by Ito et al. [166] revealed that increased serum concentration of alpha- and beta-carotene and lycopene was directly proportional to fewer ratios of cardiovascular death [166]. Bechor et al. [167] showed that 9-*cis*-β-carotene from the diet accumulates in peritoneal macrophages and increases cholesterol efflux to high-density lipoprotein (HDL) and hitherto inhibits atherosclerosis. The mechanisms of action of reported anti-CVD activities of carotenoids are not unconnected to their inhibitions of the Nrf2/Keap1, NF-κB, and MAPK signaling pathways and their ability to increase cholesterol efflux to HDL [168].

## 5. Carotenoids’ Pharmacokinetics, Dosage, and Toxicity

Carotenoids are absorbed similarly to lipids and transported into the liver via the lymphatic system. The absorption of carotenoids is a function of the diet. A diet rich in cholesterol increases the absorption of carotenoids, while a diet low in cholesterol reduces the absorption of carotenoids [168]. The bioavailability of trans-carotenoids is better than cis-carotenoids. The elimination of carotenoids from the body varies. Β-carotene takes 5–7 days, while lycopene takes 2–3 days to be eliminated. Astaxanthin‘s half-life (t_½_) is 16 h [168]. Carotenoids are well tolerated with minimal side effects [168]; carotenoids are also non-toxic. However, a high dose of canthaxanthin can cause retinopathy [169]. Also, a high intake of a β-carotene (20–30 mg/day) supplement is associated with lung cancer in smokers [170]. A dose of 15 mg (for adults) and 6 mg (for teenagers) of β-carotene is recommended as a supplement when necessary [171]. There is currently no recommended daily dose for lycopene. However, a dose between 8–21 mg per day for an adult seems beneficial [172]. A daily dose of 8 mg of astaxanthin is recommended and regarded as safe, but consumption of about 50 mg/kg of astaxanthin per day leads to pigmentation of the skin [168].

## 6. Carotenoids Absorption and Metabolism

Carotenoid absorption and bioconversion occur in the intestine. Intestinal micelles contain carotenoids, bile salts, cholesterol, fatty acids, monoacylglycerides, and phospholipids. The bioaccessibility and bioavailability of carotenoids are determined by hydrophobicity and postprandial levels [173,174]. The polar ionone rings contribute to the bioavailability of carotenoids [175]. The geometric isomerism of carotenoids also affects their bioavailability. When heated, the trans-form isomerizes to the cis-form, which affects the cartotenoids’ bioavailability [176,177]. Factors such as food matrix, formulation, food processing, and antioxidants also affect the bioavailability of carotenoids [178]. Intestinal absorptive cells (enterocytes) that traverse the membrane bilayers take up carotenoids from mixed micelles for processing in the brush border cells. Class B scavenger receptors (SR-B1) enhance carotenoid uptake from mixed micelles [179], and the absorption of carotenoids from mixed micelles is linked to cluster determinant 36 (CD36) proteins [180]. Chylomicrons absorb carotenoids and other dietary lipids before they are released and reach the lymphatic system. Retinaldehyde, the main conversion product, is transformed into retinoic acid, retinol, and retinyl esters [179]. According to Olson and Hayaishi [181] and Goodman and Huang [182], two molecules of retinaldehyde are produced when β-carotene is uniformly split at the C15, C15′ double bond. Carotenoids cleavage dioxygenases (CCDs) are responsible for this cleavage [183,184,185]. CCDs are classified into two types: β-carotene-15,15′-dioxygenase (BCO1) and β-carotene-9,10′-dioxygenase (BCO2). Carotenoids (provitamin A) and apocarotenoids with greater than twenty carbon chains are the substrates of BCO1 [186]. BCO2 has a broad substrate selectivity towards carotenoids. Acyclic canthaxanthin, zeaxanthin, lutein, and lycopene are all transformed by it [187]. BCO2 cleaves at positions C-9′ and C-10′, next to many ionone ring sites of carotenoids [178]. The intestine and other tissues express both BCO1 and BCO2 [188]. BCO1 and BCO2 are found in the cytosol and mitochondria, respectively. BCO1 is the key enzyme for producing vitamin A. An individual’s vitamin A status influences carotenoid absorption and metabolism [171,189].

## 7. Carotenoids as Medicines and Supplements

Carotenoids have several biological activities, ranging from antioxidants and anti-inflammatory activities to the inhibition of malignant tumor growth. The effects of carotenoid intake on the risk of developing certain diseases are extensively studied. As a result of those studies, several clinical trials on carotenoids are ongoing [190], and a handful of carotenoids are clinical drugs or supplements. Table 1 and Figure 5 list carotenoids marketed as systemic or topical formulations for a variety of conditions.

## 8. Engineering the Biosynthesis of Carotenoids

Carotenoids are high-value natural products due to their applications in the food, flavoring, pharmaceutical, and feed industries. However, the extraction and synthesis of carotenoids are costly, technically arduous, rigorous, and limited. Therefore, alternative ways of producing adequate quantities of carotenoids to meet the demand are highly desirable. Metabolic engineering techniques and the use of plants and microorganisms as cell factories for specialty carotenoids appear to be the future of carotenoid biosynthesis. Metabolic engineering can enhance carbon flux from substrates to target compounds while limiting the flow to unnecessary products. The carotenoid biosynthetic pathway can be put into four metabolism units: (i) central carbon metabolism, (ii) cofactor metabolism, (iii) isoprene supplement metabolism, and (iv) carotenoid biosynthesis. Metabolic engineering ensures an increased and balanced carbon flux into all four units for the overproduction of carotenoids.

Exploring the central carbon metabolism

The precursors of mevalonate (MVA) and 2-C-methyl-D-erythritol-4-phosphate (MEP) pathways, acetyl-CoA, glycerol-3-phosphate (G3P), and pyruvate are not limited to these pathways but are also important substrates for other pathways. Therefore, to increase the overproduction of carotenoids, it is imperative to reduce the by-products that utilize the biosynthetic precursors [191]. Carotenoid production can be upregulated via a knock-out of the gene encoding the E1 subunit of 2-oxoglutarate dehydrogenase, SucA, thereby decreasing acetyl-CoA consumption by the tricarboxylic acid (TCA) cycle and increasing more flux of acetyl-CoA into the MVA pathway [192]. The MEP pathway is initiated by two precursors, G3P and pyruvate, and carotenoid production is reduced by limiting one of these precursors. However, G3P is mainly converted to pyruvate, and the flux is skewed toward it. Diverting the flux from pyruvate to G3P by inhibiting glyceraldehyde 3-phosphate dehydrogenase (GapA) or overexpressing phosphoenolpyruvate (PEP) synthase (Pps) has been shown to increase carotenoid production in *E. coli* by 1.5-fold and 5-fold, respectively [193,194].

ii.Exploring co-factor metabolism

ATP and NADPH, cofactors of the MVA and MEP pathways, play key roles in carotenoid biosynthesis. The pentose phosphate pathway and the tricarboxylic acid (TCA) cycle are the major producers of NADH. Therefore, a good strategy for ensuring the overproduction of NADPH for carotenoid biosynthesis is to explore the pentose phosphate pathway and the TCA cycle. Several schemes can be used to accomplish this, including the overexpression of glucose-6-phosphate dehydrogenase (Zwf1), transketolase I (TktA), and transaldolase B. (talB) [195]. Carotenoid production and NAD(P)H supply can also be improved by overexpressing malate dehydrogenase (Mdh) or citrate synthase (GltA), α-ketoglutarate dehydrogenase (SucAB), and succinate dehydrogenase (SdhABCD) in the TCA cycle [196]. Zhao et al. [197] demonstrated that an overexpression of Glucose-6- phosphate dehydrogenase (Zwf1, NADH kinase (Pos5), and NADPH-supplying sources enhance carotenoid biosynthesis in recombinant *Saccharomyces cerevisiae.* Also, modulation of the expression of NADH-quinone oxidoreductase, cytochrome bd-I oxidase, cytochrome bo oxidase, and ATP synthase enhanced β-carotene production by 120% [196];

iii.Exploring isoprene metabolism

A key precursor for increasing carotenoid production is isopentenyl diphosphate (IPP) [198]. IPP is synthesized from the MVA and MEP pathways. Directed evolution and the overexpression of mevalonate kinase (MK) can be used to boost IPP downstream product yields [199]. Several feedback-resistant MKs might also be expressed in microorganisms to increase carotenoid production [200]. In addition, feedback inhibition of 3-hydroxy-3-methylglutaryl-coenzyme A reductase can restrict carotenoid production (HMGR1). This limitation can be addressed by overexpressing truncated HMGR1 (tHMGR1) [201]. The supply of IPP precursors can also be increased by optimizing the expression of the upper MVA pathway with tunable intergenic regions (TIGRs). Tunable expression of the lower MVA pathway was also shown by Shen et al. [202] to increase the production of zeaxanthin. Galactose-regulated promoters attached to MVA pathway genes were shown to reduce toxic intermediate generation and increase IPP supply for tunable expression of the entire MVA [203]. Furthermore, ROX1, a transcriptional regulator that inhibits genes involved in the MVA pathway, can be deleted to increase IPP supply [204];

iv.Carotenoid biosynthesis in Microorganisms

Microbes, such as *Escherichia coli* and *Saccharomyces cerevisiae*, sometimes lack the genes to synthesize desired natural products. Because heterologous proteins are expressed differently in different hosts [205], gene sources that generate carotenoid biosynthesis pathways can be chosen carefully to accomplish heterologous gene cluster expression. Yoon et al. [206] demonstrated in *E. coli* that switching the crtE/B/I genes from *Pantoea ananatis* to *Pantoea agglomerans* nearly doubles lycopene production. Single genes encoding proteins with phytoene synthase and lycopene cyclase activities from many species have been functionally studied and utilized for carotenoid biosynthesis [207,208]. Because the rate-limiting step in carotenoid biosynthesis is geranylgeranyl diphosphate (GGPP) synthase’s conversion of dimethylallyl diphosphate (DMAPP) to geranylgeranyl diphosphate (GGPP), the production of carotenoids has been increased by the directed evolution of GGPP synthase [209], which increases GGPP synthesis. Farnesyl pyrophosphate (FPP) synthase mutants can increase GGPP synthesis and provide downstream products [210,211]. Carotenoid production can also be improved by increasing gene copy numbers for key enzymes since protein expression levels are typically related to gene copy numbers [212]. Wu et al. [213] found that the overexpression of glycerol-3-phosphate O-acyltransferase and 1-acyl-sn-glycerol-3-phosphate acyltransferase increases membrane synthesis and improves β-carotene levels. To fully maximize the potential of microbes in carotenoid production, it is crucial to have a complete, accurate genome sequence before mining the genetic features of microbes to decode their functional activities [214]. High-throughput sequencing technologies are essential for generating such genomic data. Third-generation sequencing (TGS) PacBio single-molecule real-time (SMRT) technology and Illumina sequencing, a traditional next-generation sequencing technology (TNGS), provide the technology necessary to accomplish it [215]. Ma et al. [216] have used a combination of PacBio-Illumina whole-genome sequencing techniques in *Sphingomonas sp*. ATCC 55669 to identify a new astaxanthin producer and provided additional insights into the biosynthetic components for astaxanthin bioengineering. A systematic study of the whole genome sequencing of *R. glutinis* X-20 by Bo et al. [217] revealed that the crtI and crtYB genes are involved in carotenoid biosynthesis;

v.Plants as cell factories for specialty carotenoids

The global carotenoids market was estimated to be about $1.44 billion in 2019, and it is projected to reach $1.84 billion by 2027 [218]. The use of carotenoids in the food, supplement, and pharmaceutical industries will continue to increase. Using plant cells as carotenoid factories is one metabolic engineering technique to meet the demand for carotenoids. As shown by Ralley et al. [219], the introduction of β-carotene ketolase (CrtW) and β-carotene hydroxylase (CrtZ) genes into tobacco resulted in the accumulation of high levels of ketocarotenoids in the nectar. Also, astaxanthin overproduction was observed in the root of carrot by introducing a β-carotene ketolase gene (CrtO) isolated from the *alga Haematococcus pluvialis* [220]. Crocins, non-endogenous apocarotenoids, were also produced in *Nicotiana glauca* leaves and petals by a constitutive expression of the BdCCD4.1 gene [221]. Lastly, the production of significant levels of β-carotene in rice through metabolic engineering and its pending introduction to the food supply chain provides an excellent example of the role of plants in addressing nutrient deficiency using staple food crops [222].

## 9. Conclusions

Carotenoids are isoprenoids responsible for the different colors of plants, fruits, and vegetables. Carotenoids play significant roles in human health, and there are many ongoing pharmacological and drug discovery studies on the antineoplastic, antidiabetic, anti-CVD, and anti-aging actions of carotenoids. Some carotenoids are used to treat skin and ocular diseases and leukemia, and the list of conditions treated with carotenoids could increase in the near future. The metabolic engineering of carotenoid biosynthesis holds great promise for the efficient production of carotenoids for human consumption and drugs. Further laboratory and clinical studies are needed to understand better the mechanistic details of carotenoids’ role in human health.

## Figures and Tables

**Figure 1 molecules-27-06005-f001:**
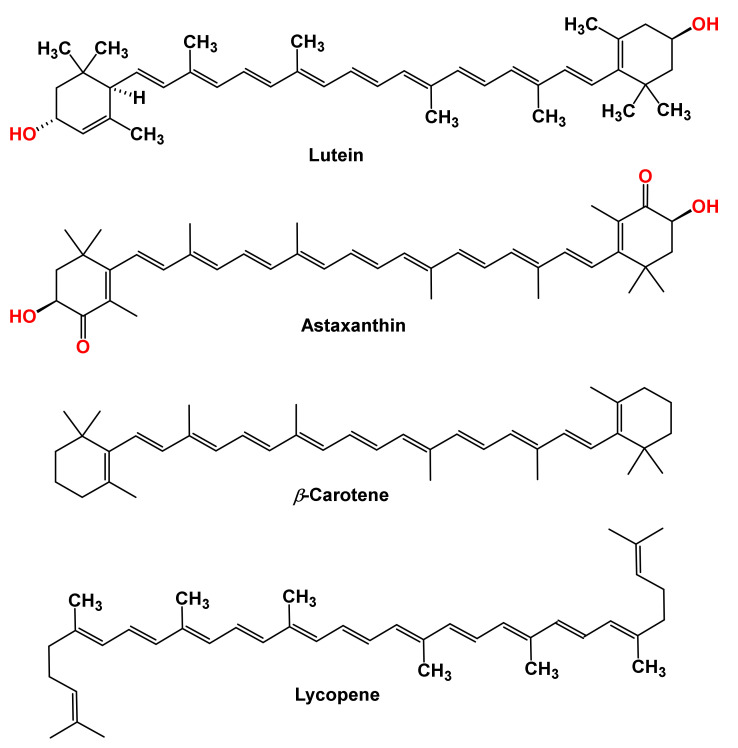
Structures of common carotenoids.

**Figure 2 molecules-27-06005-f002:**
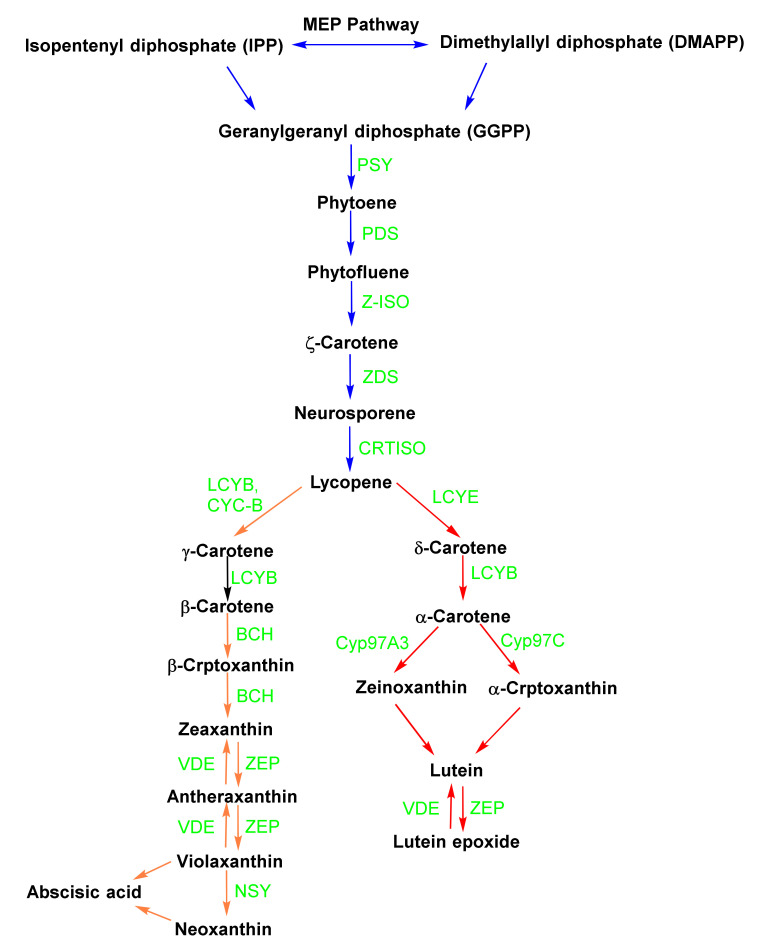
The carotenoid biosynthetic pathway in plants. PSY: phytoene synthase; PDS: phytoene desaturase; Z-ISO: ζ-carotene isomerase; ZDS: ζ-carotene desaturase; CRTISO: carotene isomerase; LCYE: lycopene ε-cyclase; LCYB: lycopene β-cyclase; CYC-B: chromoplast-specific lycopene β-cyclase; BCH: β-carotene hydroxylase; ZEP: Zeaxanthin epoxide; VDE: violaxanthin epoxide; NSY: neoxanthin synthase; CYP97A3: cytochrome P450-type β-hydroxylase; and CYP97C1: cytochrome P450-type ε-hydroxylase.

**Figure 3 molecules-27-06005-f003:**
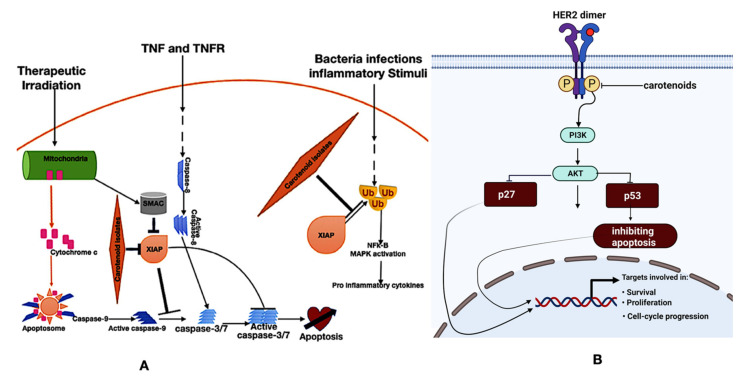
Mechanisms of carotenoids from *Spondias mombin*: (**A**) antagonism of XIAP regulation of apoptotic and inflammatory signaling and (**B**) inhibition of the kinase domain in the HER2 signaling pathway.

**Figure 4 molecules-27-06005-f004:**
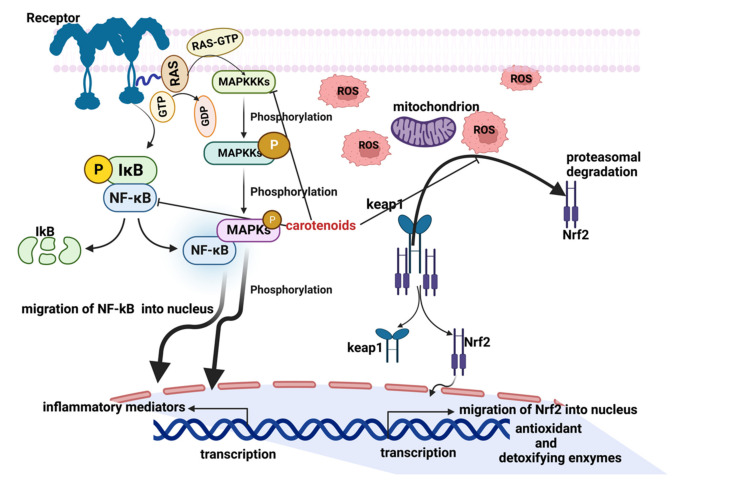
Mechanisms of the anti-diabetic effects of carotenoids. During oxidative stress, IκB is phosphorylated by IKK, resulting in the release of NF-κB. NF-κB is translocated to the nucleus, and the transcription of inflammatory cytokines can commence. Carotenoids inactivate the NF-κB pathway. Carotenoids inhibit the degradation of Nrf2 and interact with Keap1. Nrf2 migrates to the nucleus, and the transcription of antioxidant and detoxifying enzymes can start. Carotenoids inhibit MAPK signaling from reaching the nucleus [101,102,103].

**Figure 5 molecules-27-06005-f005:**
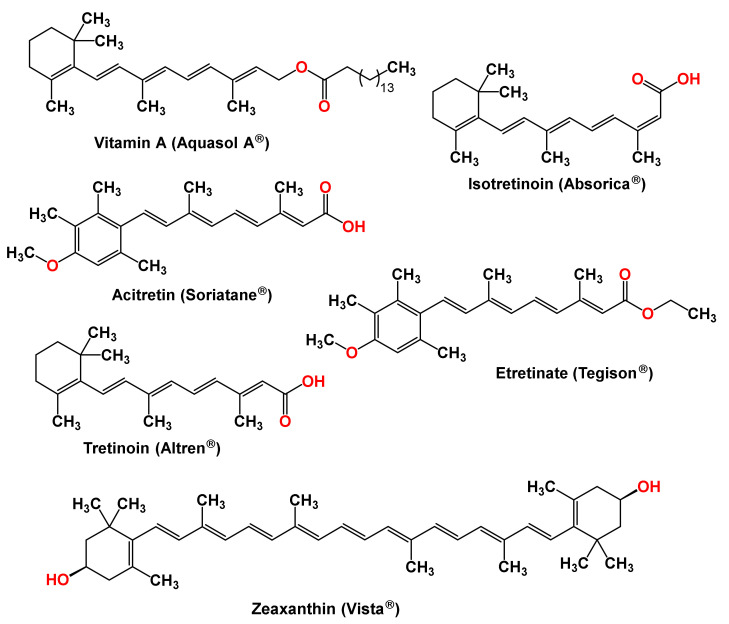
Structures of marketed carotenoid-based drugs and supplements.

**Table 1 molecules-27-06005-t001:** List of marketed carotenoid-based drugs and supplements.

Active Ingredients	Brand Name	Country	Route of Administration	Disease/Indication
**Zeaxanthin**	Vista Advanced AREDS2 Formula	USA	Oral	Age—Related Macular Degeneration (AMD)/Cataract, Glaucoma, NPDR—Non-Proliferative Diabetic Retinopathy/Type 2 Diabetes Mellitus
**Vitamin A Palmitate**	Aquasol A	USA	Intramuscular	Preventing cataracts, Retinitis pigmentosa, Diarrhea in pregnant women Vitamin deficiency
	Infuvite	Canada	Intravenous	
	Infuvite Pediatric, Mvi Pediatric	USA	Intravenous	
	Mvc-fluoride	USA	Oral	
***β*-carotene**	Pregvit	USA	Oral	Prenatal/postpartum vitamin and mineral supplements, reduction of photosensitivity in patients with erythropoietic protoporphyria and other photosensitivity diseases
**Tretinoin**	Altreno, Atralin, Avita, Renova	USA, Canada	Topical, Oral	Treat acne vulgaris, certain types of promyelocytic leukemia, and fine wrinkles.
**Etretinate**	Tegison	USA	Oral	Used to treat severe psoriasis
**Acitretin**	Soriatane	USA, Canada	Oral	For the treatment of psoriasis
**Isotretinoin**	Absorica, Accutane, Amnesteem, Claravis, Clarus, Epuris, Myorisan, Sotret, Zenatane	USA	Oral	Treat severe recalcitrant nodular acne
**Lycopene**	IQQU Advanced Sunscreen SPF 50	USA	Topical	Defends against skin sensitivity, including dryness, irritation, roughness, tightness, and a weakened skin barrier.
	Omega-3 Rx Complete	USA	Oral	Adjunct to diet to reduce triglyceride (TG) levels in adult patients with severe (≥500 mg/dL) hypertriglyceridemia (HTG)

## Data Availability

Not applicable.

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
