# Peer review of "Carotenoids in Drug Discovery and Medicine: Pathways and Molecular Targets Implicated in Human Diseases"

_molecules, 2022, doi:10.3390/molecules27186005_

Round 1

Reviewer 1 Report

My comments are as below:

The abstract should include information summarising the authors' crucial findings in the form of numerical data.

The second paragraph of this manuscript's introduction needs to be significantly fleshed out and provide a clear hypothesis.

In general, there is unnecessary information repetition.

Ligand figures need to be edited; they are poorly written.

More details and references to relevant and related literature should be included in the discussion.

Author Response

Reviewer I Comments

Response

The abstract should include information summarizing the authors' crucial findings in the form of numerical data.

The abstract includes information summarizing the content of the review article.  

The second paragraph of this manuscript's introduction needs to be significantly fleshed out and provide a clear hypothesis.

The second paragraph of the manuscript's introduction provides a good overview of the content of the review article

In general, there is unnecessary information repetition.

All unnecessary information and repetition have been removed from the manuscript.

Ligand figures need to be edited; they are poorly written.

The figures and their legends are appropriately presented.

More details and references to relevant and related literature should be included in the discussion.

References relevant to the review are appropriately cited.

Reviewer 2 Report

Title: Carotenoids in Drug Discovery and Medicine: Pathways and Molecular Targets Implicated in Human Diseases

Authors: Damilohun Samuel Metibemu, Ifedayo Victor Ogungbe

Comments:

This review paper describes carotenoids from their extraction to their use in various human diseases. The results compiled are not bad, but this work in its present form is not suitable for publication. The English writing style is poor and the review remains thematically superficial. It needs massive revision in terms of content, structure, and language.

Author Response

Reviewer II Comments

Response

This review paper describes carotenoids from their extraction to their use in various human diseases. The results compiled are not bad, but this work in its present form is not suitable for publication. The English writing style is poor and the review remains thematically superficial. It needs massive revision in terms of content, structure, and language.

a.       We agree that the English writing style could be improved and enhanced. We have done that. Thank you for your suggestions.

b.      The content is undoubtedly thematic and provides an up-to-date summary of carotenoids’ use and importance in human health.

Round 2

Reviewer 1 Report

The authors have significantly improved the manuscript, it can be accepted after a careful English language check.

Reviewer 2 Report

The authors have significantly improved the manuscript.